# Immunotherapy Combined with Chemotherapy in the First-Line Treatment of Advanced Gastric Cancer: Systematic Review and Bayesian Network Meta-Analysis Based on Specific PD-L1 CPS

**DOI:** 10.3390/curroncol32020112

**Published:** 2025-02-16

**Authors:** Wenwei Zhang, Kaibo Guo, Song Zheng

**Affiliations:** 1Department of Oncology, The Fourth School of Clinical Medicine, Zhejiang Chinese Medical University, Hangzhou 310006, China; 13212831715@163.com; 2Department of Oncology, Hangzhou First People’s Hospital, Hangzhou 310006, China; guokaibo@zcmu.edu.cn; 3Zhejiang University School of Medicine, Hangzhou 310006, China; 4Key Laboratory of Clinical Cancer Pharmacology & Toxicology Research of Zhejiang Province, Affiliated Hangzhou First People’s Hospital, Westlake University, Hangzhou 310006, China

**Keywords:** immunotherapy, chemotherapy, advanced gastric cancer, network meta-analysis

## Abstract

**Objective:** To compare the efficacy and safety of immunotherapy combined with chemotherapy as the first-line treatment for advanced gastric cancer. **Data Sources**: Phase III randomised controlled trials were searched from PubMed, Embase, Web of Science, Cochrane Library, and ClinicalTrials databases, and several international conference databases, from inception to 15 November 2024. **Results**: A total of eight eligible trials involved 7898 patients and eight treatments. The network meta-analysis showed that cadonilimab plus chemotherapy was the most superior treatment in improving overall survival (versus conventional chemotherapy, hazard ratio 0.62, 95% credible interval 0.50 to 0.78) and progression-free survival (0.53, 0.43 to 0.65), and consistency of results were observed in specific PD-L1 combined positive score groups. All immune checkpoint inhibitors combined with chemotherapy improved patient prognosis, but nivolumab plus chemotherapy may lead to an increase in grade 3 or higher adverse events (odds ratio 1.68, 95% credible interval 1.04 to 2.54), and the toxicity of cadonilimab plus chemotherapy was more likely to force patients to discontinue treatment. **Conclusions**: These results showed that cadonilimab plus chemotherapy had the best overall survival and progression-free survival benefits for advanced gastric cancer patients with HER-2 negative, and was preferentially recommended to patients with positive PD-L1 CPS.

## 1. Introduction

In 2022, more than 968,000 people worldwide were newly diagnosed with gastric cancer, making it the fifth most common type of cancer, just behind lung, female breast, colorectum, and prostate cancer. In the same year, close to 660,000 people died from gastric cancer [1]. Drug treatment for advanced gastric cancer (AGC) was mainly chemotherapy, usually using platinum and fluoropyrimidine as perioperative chemotherapy and palliative chemotherapy. The prognosis of advanced gastric cancer is also influenced by a number of factors, including gender, age, race, tumour-related factors (tumour location, histological type, depth of infiltration and metastasis) and PD-L1 expression [2,3,4,5].

Immune checkpoint inhibitors (ICIs), which inhibit coinhibitory molecules such as cytotoxic T lymphocyte-associated antigen-4 (CTLA-4), programmed cell death protein-1 (PD-1), and the related programmed death-ligand 1 (PD-L1) [6,7,8], often administered to gastric cancer patients include pembrolizumab (anti-PD-1 monoclonal antibody) and cadonilimab (anti-PD-1/CTLA-4 bispecific antibody), etc. Clinical data have revealed that nivolumab combined with platinum-based or/and fluoropyrimidine-based chemotherapy can effectively improve the survival of AGC patients [9,10]. The favourable outcomes of the KEYNOTE-859 indicated that the combination of pembrolizumab and platinum-based chemotherapy might be a promising choice for the treatment of advanced gastric cancer, which challenged the results of the previously published KEYNOTE-062 [11,12]. In 2024, the V.2 NCCN guidelines reaffirmed using nivolumab and pembrolizumab in the first-line treatment of gastric cancer with PD-L1 CPS ≥ 5 and CPS ≥ 1, respectively. The latter was particularly recommended for patients with PD-L1 CPS ≥ 10. Furthermore, ORIENT-16 and COMPASSION-15 conducted in Asia revealed that sintilimab and cadonilimab showed significant superiority for patients with PD-L1 CPS ≥ 5 or 10 [13,14].

With these developments, questions have arisen concerning the relative efficacy and safety between the various first-line treatments. Multiple randomized controlled trials and conventional pairwise meta-analyses have been performed to evaluate the therapeutic effectiveness and safety profiles of first-line therapies for AGC, though these analyses have been restricted to direct comparative methodologies. Previous network meta-analyses only incompletely compared treatments but have yet to include the latest alternative treatments or available trials, and also lack specificity in terms of accurate treatment profiles for patients with specific PD-L1 CPS. A network meta-analysis of RCTs was performed to investigate the relative efficacy and toxicity of immune and chemotherapy combination (IC combination) as first-line treatments for AGC patients with specific PD-L1 CPS. Then, subgroup analyses were performed based on ethnicity, chemotherapy regimen, primary site of tumour, liver metastases, microsatellite instability, and Lauren typing. Furthermore, a Bayesian framework was utilized to rank treatments within each subgroup.

## 2. Materials and Methods

Our researchers performed this network meta-analysis based on Preferred Reporting Items for Systematic Reviews and Meta-Analysis extension statements [15,16] (Appendix A).

### 2.1. Data Sources and Searches

We systematically searched the PubMed, Embase, Web of Science, Cochrane Library, and ClinicalTrials.gov databases to include relevant studies published in English between January 2013 and May 2024. To include complete and updated outcomes, abstracts and presentations of ongoing randomized controlled trials on advanced gastric cancer from several critical international conferences (ASCO, ESMO, etc.) from 2021 to 2024 were inspected. The keywords for the literature search were “advanced gastric cancer, randomized clinical trial, immunotherapy, PD-1, PD-L1, CTLA-4” (Appendix A).

### 2.2. Study Selection

Published and unpublished phase III RCTs that met the following criteria were included:Trials that enrolled patients with histologically or cytologically confirmed unresectable locally advanced or metastatic gastric or gastroesophageal junction adenocarcinoma.RCTs that compared any two or more different arms of first-line treatments for patients with advanced gastric cancer.RCTs that used IC combination as first-line treatment settings.Phase III trials that reported on at least two of the following clinical outcome measures:Overall survival (OS), operationally defined as the temporal interval spanning from treatment randomization to all-cause mortalityProgression-free survival (PFS), defined as the time interval from randomization until the first documented occurrence of disease progression (locoregional or distant) or death from any cause.Toxicity regarding treatment-related adverse events (TRAEs) grade greater than or equal to 3 was graded according to the National Cancer Institute Common Terminology Criteria for adverse eventsAdverse events (AEs) leading to treatment discontinuation, defined as any AEs leading to discontinuation of study due to the discontinuation of one or more drug

Trials meeting the above criteria were included in this study, and the exclusion criteria were as follows:Trials reporting results for patients with HER-2 positive gastric adenocarcinoma (Patients receiving or having received anti-HER2 targeted therapy)Trials in which ICIs were used as maintenance or neoadjuvant treatments or as sequential treatments with chemotherapy

### 2.3. Data Extraction and Risk of Bias Assessment

Detailed data such as study ID, patient characteristics, treatments, and outcomes were extracted and entered into an electronic spreadsheet (Table 1). Ideally, the independent review facility should extract survival-related data to avoid potential assessment bias caused by the investigators. Risk of bias assessment for individual clinical studies was conducted using the Cochrane Risk of Bias Tool, encompassing key domains: randomization sequence generation, allocation concealment, blinding of participants and investigators, blinding of outcome assessors, completeness of outcome data, selective reporting of results, and identification of other potential bias sources [17]. The items were classified as having a low, high or unclear risk of bias. Two researchers (WZ and KG) independently extracted the data and assessed the risk of bias, and any discrepancies were resolved by consensus and arbitration by a panel of adjudicators (SZ).

### 2.4. Data Synthesis and Statistical Analysis

We obtained indirect evidence by processing and analysing direct evidence to compare the efficacy and safety of various treatments. The reported results included hazard ratios for survival outcomes (overall survival and progression-free survival) and odd ratios for binary outcomes (grade ≥ 3 TRAEs and AEs leading to treatment discontinuation), as corresponding 95% confidence intervals. The primary outcomes were overall survival and progression-free survival, and the secondary outcomes were grade ≥ 3 TRAEs and AEs leading to treatment discontinuation.

We used Stata software (version 17.0) to generate network plots of the specific outcomes for the various subgroups to clarify which treatments were directly or indirectly compared in the included studies and to illustrate the geometry of the network plots [18]. The heterogeneity of the studies was assessed using the Q test and I2 and represented in forest plots [19]. The difference is considered statistically significant when the *p*-value is less than 0.05. Heterogeneity was considered low, moderate or high when the I2 estimates were below 25%, between 25% and 50%, and above 50%, respectively.

This network meta-analysis was conducted using the Markov chain Monte Carlo simulation technique in OpenBUGS (version 3.2.3) under a Bayesian framework. For each study endpoint, three independent Markov chains were established using a fixed-effect consistency model [20], and 100,000 sample iterations were generated with 40,000 burn-ins and a thinning interval of 10. The convergence of the calculated model was estimated using the history feature [21]. The process aims to yield a posteriori distribution and to use trace plots and the Brooks–Gelman–Rubin diagnostic to estimate and visually assess the convergence of the model iterations (Appendix A). Provided that there are minimally informative priors, credible intervals can be interpreted in the same way as conventional confidence intervals. Similarity, consistency, and transitivity are the three essential assumptions of network meta-analysis. The existence of transitivity makes it possible to make indirect comparisons between treatments that do not have direct comparisons [22]. Then, the global inconsistency is estimated by comparing the fit of consistency and inconsistency models [23,24].

To make the results of this analysis more robust and reliable, we also planned the sensitivity analyses. Due to an increase in early deaths, the data monitoring committee for CheckMate 649 recommended that enrolment in the nivolumab plus ipilimumab arm be ended early. We performed the sensitivity analysis after excluding the nivolumab plus ipilimumab group in CheckMate 649 [9]. In addition, a network meta-analysis based on Asian patients was also completed, but we only analysed the overall survival due to data acquisition limitations. Network meta-analysis based on a chemotherapy regimen was also completed for overall survival.

## 3. Results

### 3.1. Systematic Review and Characteristics

A total of 5308 records were retrieved from the database and international conferences. After the screening process of reading the title and abstract, 79 articles were retained, their full text was browsed, and eight eligible trials were identified (Figure 1). Eight randomized controlled trials involving 7898 patients were included in the review. Chemotherapy was used as induction therapy in the JAVELIN Gastric 100 [25]. Considering the inconsistency of the starting point of the survival time measure between JAVELIN Gastric 100 and other eligible trials, it was not included in our consideration. Treatments eligible for network meta-analysis are tislelizumab plus chemotherapy (tisl-chemo), nivolumab plus chemotherapy (nivo-chemo), nivolumab plus ipilimumab (nivo-ipi), pembrolizumab plus chemotherapy (pemb-chemo), sintilimab plus chemotherapy (sint-chemo), cadonilimab plus chemotherapy (cado-chemo), sugemalimab plus chemotherapy (suge-chemo), and chemotherapy (chemo) [9,10,11,12,13,14,26,27,28]. These networks are shown in Figure 2. The basic characteristics of the patients are listed in Table 1. A detailed risk assessment of bias is summarised in Appendix A.

### 3.2. Comparisons of OS and PFS in ITT Population

Network meta-analysis based on intent-to-treat (without PD-L1 selection, ITT) population included six studies involving seven treatment methods for overall survival and progression-free survival (Figure 2A), and the data referenced were from trials with longer follow-up than previously published.

Regarding OS (Figure 3A), patients receiving all IC combinations were more probably to gain greater OS benefits than those receiving conventional chemotherapy. Cado-chemo yielded a favourable benefit (versus nivo-ipi (hazard ratio 0.68, 95% credible interval 0.52 to 0.90), nivo-chemo (0.76, 0.60 to 0.97), tisl-chemo (0.78, 0.60 to 1.00), chemo (0.62, 0.50 to 0.78)). There was no significant statistical difference between pemb-chemo, sint-chemo, and cado-chemo.

Regarding PFS (Figure 3A), the PFS benefits of all IC combinations compared with chemotherapy were consistent with the results of OS. Cado-chemo, compared with other treatments, showed a better PFS benefit (versus pemb-chemo (hazard ratio 0.69, 95% credible interval 0.56 to 0.88), nivo-ipi (0.32, 0.25 to 0.41), nivo-chemo (0.68, 0.55 to 0.85), tisl-chemo (0.68, 0.53 to 0.86), chemo (0.53, 0.43 to 0.65)), but did not obtain benefit compared with sint-chemo (0.83, 0.63 to 1.10).

### 3.3. Comparisons of OS in Asian Patients

The incidence of gastric cancer is higher in Asia than in most other regions, and most clinical trials have separate analyses of the Asian population [1,29]. We performed a separate analysis for Asian patients, and only a comparison of overall survival was feasible due to limitations in data availability. Six trials were included in this analysis, involving six treatments (Figure 2B). As shown in Figure 3C, all IC combinations produced greater OS benefits than traditional chemotherapy. Cado-chemo provided greater survival benefits than tisl-chemo (hazard ratio 0.75, 95% credible interval 0.57 to 0.98) and nivo-chemo (0.76, 0.58 to 0.99), but no statistically significant difference compared to pemb-chemo (0.87, 0.65 to 1.18) and sint-chemo (0.80, 0.60 to 1.08).

### 3.4. Comparisons of OS in Patients Receiving Chemotherapy with XELOX

At present, oxaliplatin combined with capecitabine (XELOX) has been the standard first-line chemotherapy regimen. In this network meta-analysis, we included six trials involving six treatments to compare overall survival between several ICIs combined with the same chemotherapy regimen (Figure 2C). Similar to the results above, the IC combinations showed a better OS benefit compared to conventional chemotherapy (Figure 3C). However, in the comparisons of the various IC combinations, only cado-chemo and nivo-chemo showed a statistically significant difference (hazard ratio 0.76, 95% credible interval 0.58 to 0.99).

### 3.5. Comparisons of OS Based on PD-L1 CPS

We performed a network meta-analysis to compare overall survival based on specific PD-L1 CPS (Figure 2E–G) and calculated the ranking of various treatments according to the Bayesian method (Appendix A).

For PD-L1 CPS ≥ 1 patients (Figure 3D), the significant differences in cado-chemo versus nivo-ipi (hazard ratio 0.67, 95% credible interval 0.49 to 0.92), and conventional chemotherapy (0.58, 0.44 to 0.75), and the marginal difference in cado-chemo versus pemb-chemo (0.75, 0.56 to 0.99), and nivo-chemo (0.78, 0.59 to 1.05) were in favour of cado-chemo as the best treatment among the comparable IC combinations. In comparisons for other IC combinations, most hazard ratios were close to 1, with no significant difference observed. We observed that all IC combinations conferred a survival benefit over conventional chemotherapy, which is consistent with the findings above.

For PD-L1 CPS ≥ 5 patients (Figure 3D), all IC combinations provided significant differences in hazard ratios compared to conventional chemotherapy, chemo versus cado-chemo (hazard ratio 1.86, 95% credible interval 1.31 to 2.64), versus sint-chemo (1.52, 1.16 to 1.99), versus suge-chemo (1.34, 1.09 to 1.64).

For PD-L1 CPS ≥ 10 patients (Figure 3E), we also found that IC combinations reduced the hazard ratios more with the PD-L1-CPS increasing, suge-chemo versus chemo (hazard ratio 0.65, 95% credible interval 0.49 to 0.86), cado-chemo versus chemo (0.51, 0.33 to 0.80), sint-chemo versus chemo (0.56, 0.41 to 0.77), pemb-chemo versus chemo (0.68, 0.58 to 0.81), nivo-chemo versus chemo (0.66, 0.56 to 0.78). This would seem to argue that as the patient’s PD-L1 CPS increases, immunotherapy combined with chemotherapy was more appropriate as the preferred treatment option.

### 3.6. Comparisons of Safety and Toxicity

Safety and toxicity were determined according to treatment-related adverse events grade greater than or equal to 3 and adverse events leading to treatment discontinuation. Network meta-analysis included eight trials involving eight treatments (Figure 2D).

We did not find any statistically significant safety signals in the comparisons between IC combinations, including TRAEs grade greater than or equal to 3 and AEs leading to treatment discontinuation. Compared with conventional chemotherapy, nivo-chemo was more likely to result in grade 3 or higher TRAEs, and the toxicity caused by cado-chemo more often forces patients to discontinue treatment early (Figure 3B).

This article selected 12 adverse events with high clinical significance for reference. A heat map was created (Appendix A) for the probability of adverse events occurrence based on eligible trials [9,10,11,12,13,14,26,27,28]. It should be noted that the results of the heat map cannot explain the variability of the toxicity profile between individual IC combinations, but they can illustrate the extent of damage to different organs from the same treatment. The damage to the bone marrow caused by IC combinations was greater than to other organs such as the liver, kidneys, and thyroid gland. The most common AEs with cado-chemo and sint-chemo were platelet count decreased and neutrophil count decreased. The adverse reaction most likely to occur in patients treated with suge-chemo and pemb-chemo was anaemia, and we also observed that the spectrum of adverse effects in tisl-chemo and nivo-chemo was somewhat balanced.

### 3.7. Subgroup Analysis

We performed subgroup analyses of overall survival based on the microsatellite instability status, primary tumour site, liver metastasis status, disease status, and Lauren classification and calculated the probability of each treatment ranking first in each subgroup (Figure 4).

Our analytical results consistently demonstrated that regardless of whether the primary tumour was located in the stomach or at the gastroesophageal junction, cado-chemo exhibited the highest probability of being ranked first in therapeutic efficacy. In addition to cado-chemo, sint-chemo may confer greater therapeutic benefit in gastric tumours, while tisl-chemo demonstrates more pronounced efficacy in gastroesophageal junction tumours. In patients with liver metastasis, cado-chemo demonstrated the highest likelihood of first-line therapeutic superiority, whereas sint-chemo appeared to confer a comparative therapeutic advantage in cases without liver involvement. We conducted stratified analyses based on gastric cancer disease status. The results demonstrated that cado-chemo remains the most clinically advantageous regimen for metastatic gastric cancer, while revealing no statistically significant difference between IC combinations and chemotherapy alone in locally advanced disease. In the stratified analysis based on Lauren classification, nivo-chemo provided greater benefits than pembrolizumab combined with chemotherapy in gastric cancer of the intestinal type. Conversely, the analysis results for gastric cancer of the diffuse type were the opposite. There have been many literature reports on the relationship between the microsatellite instability status and the application of immune checkpoint inhibitors [30,31]. Gastric cancer of the MSI-H type represents only a small fraction of all cases, yet our results demonstrated excellent responsiveness to immune combination therapy, particularly the nivo-ipi. In contrast, for MSS tumours, analysis had shown that the nivo-ipi did not provide as much benefit as the IC combination.

### 3.8. Rank Probabilities

The Bayesian ranking of comparable treatments in different subgroups are shown in Figure 5 (the detailed ranking results compiled in Appendix A). We found that the results of the Bayesian ranking analysis were almost in line with the pooled analyses using hazard and odds ratios. The efficacy superiority of cado-chemo was evident across several subgroups, which may be related to its ability to confer lower hazard ratios. The overall survival benefit conferred by sint-chemo was regarded as being of secondary importance, although its superiority in Asian patients was outstripped by pemb-chemo. The OS benefit that can be derived from conventional chemotherapy alone was considered minimal, and nivo-ipi was more likely to lead to disease progression. In contrast to its superior efficacy, cado-chemo was regarded as the treatment most likely to result in patients being unable to tolerate its adverse effects and thus choosing to terminate one or more drugs prematurely.

### 3.9. Heterogeneity and Inconsistency Assessment

Four feasible pairwise comparisons with heterogeneity estimates were illustrated as forest plots in Appendix A. The heterogeneity assessment showed that relatively high heterogeneity exists in comparisons regarding progression-free survival (I2 = 67.0%) and AEs leading to treatment discontinuation (I2 = 51.1%). The fit of the consistency model was similar to or better than that of the inconsistency model (Appendix A). The inconsistency between the direct and indirect estimates was supposed to be assessed by node splitting analysis, but our networks did not have a closed-loop structure, thus rendering this process unnecessary.

### 3.10. Sensitivity Analysis

Due to an increase in early deaths, the data monitoring committee for CheckMate 649 recommended that enrolment in the nivolumab plus ipilimumab arm be ended early, which was suspected to increase heterogeneity in our network meta-analysis. After excluding the treatment of nivo-ipi [9], we performed the first sensitivity analysis. With a total of 7080 patients, eight phase III trials [10,11,12,13,14,26,27,28] involving seven treatments were included in the sensitivity analysis (Appendix A). The results of the sensitivity analysis did not show relevant deviations compared to the raw network meta-analysis but showed the lowest toxicity for conventional chemotherapy (Appendix A and Appendix A).

During the heterogeneity assessment, we identified moderate heterogeneity in the analysis of PFS and AEs, leading to treatment discontinuation. A comprehensive review of trial designs, study populations, and statistical methodologies across individual trials suggested that the observed heterogeneity likely stemmed from random variations within the dataset. To address this, we conducted leave-one-out sensitivity analyses by iteratively excluding each trial (Appendix A), combined with heterogeneity evaluation results (Appendix A). This process indicated that the COMPASSION-15 trial was the primary source of heterogeneity.

Following the exclusion of this trial, the second sensitivity analysis was performed. The results demonstrated that IC combinations maintained superior efficacy across all survival-related subgroups. Regarding safety, nivo-chemo was associated with a higher likelihood of grade ≥ 3 adverse events compared to chemotherapy alone, while other IC combinations did not show statistically significant differences in safety profiles. Across all subgroup analyses, sint-chemo consistently emerged as the most efficacious IC combination, with no additional safety concerns observed compared to chemotherapy alone (Appendix A).

## 4. Discussion

### 4.1. Principal Findings

This systematic review and network meta-analysis aims to provide a comprehensive overview of the comparative efficacy and toxicity of a range of first-line treatments for AGC patients. The results of the analysis indicate that:Results from all specific groups indicated that cadonilimab, an anti-PD-1/CTLA-4 bispecific antibody, had shown the best overall survival and progression-free survival benefit when combined with conventional chemotherapy. The primary toxic effect was damage to the hematopoietic system, particularly platelet and neutrophil count decreased.Among the various PD-1/PD-L1 monoclonal antibodies, the combination of sintilimab and platinum-based chemotherapy demonstrated the most favourable overall survival and progression-free survival.In the absence of unequivocal evidence that the patient’s tumour was MSI-H, the administration of nivo-ipi as a standalone first-line treatment may result in the early progress of the disease.All IC combinations improved patient prognosis, but nivo-chemo may lead to an increase in grade 3 or higher adverse events, and the toxicity of cado-chemo was more likely to force patients to discontinue treatment with one or more drugs.

The use of immunotherapy to boost T cell activity against cancer cells, such as through the blockade of CTLA-4, PD-1 or PD-L1, has been shown to have beneficial effects in AGC patients. A combination of ICIs and chemotherapy has emerged as a highly effective immunotherapeutic strategy, with its efficacy a priori validated by its incorporation into the standard treatment protocols for numerous cancers [32]. A previous clinical study demonstrated that ipilimumab was efficacious in enhancing the therapeutic response when administered in conjunction with PD-L1 antibody in patients with advanced renal cell carcinoma, melanoma, and non-small cell lung cancer [32,33]. Nevertheless, the concurrent administration of PD-1 and CTLA-4 inhibitors has been observed to elevate the prevalence of severe immune-related adverse events [34,35]. Studies revealed that, in addition to engaging immune cells, bispecific antibodies also have a role in the delivery of payloads, as co-factor mimics, and the inhibition or activation of receptors [36]. The complementary roles of PD-1 and CTLA-4 in immune response regulation (through the processes of preventing T cell depletion and promoting T cell activation, respectively) indicate that bispecific antibodies could enhance the functionality of tumour-infiltrating T cells when compared to anti-CTLA-4 or PD-L1 antibodies alone [37]. Cadonilimab represents the inaugural anti-PD-1/CTLA-4 bispecific antibody to be employed in the treatment of advanced gastric cancer [14]. To our knowledge, this was the first ICI reported to prolong both overall survival and progression-free survival in patients with PD-L1 CPS ≤ 5. The recently reported ORIENT-16 results showed that AGC patients with CPS ≥ 5 achieved a median overall survival of 19.2 months [13], which was significantly higher than the predicted outcome reported in COMPASSION-15 [14]. However, our network meta-analysis suggested that cadonilimab combined with XELOX was more likely to provide patients with the desired therapeutic effect than sintilimab combined with XELOX. A comparison of the baseline characteristics of patients in the ORIENT-16 and COMPASSION-15 revealed that the former had a more favourable profile, with a younger median age, a lower proportion of liver metastases, and a higher proportion of high PD-L1 expression. In addition, the maintenance treatments in the COMPASSION-15 were cadonilimab single agent and placebo, respectively, while in the ORIENT-16 were sintilimab combined with capecitabine and placebo combined with capecitabine, respectively. These may explain the results of the current study favouring cado-chemo over sint-chemo and affirm the necessity of using hazard ratio to measure survival benefit rather than median survival. Furthermore, we observed variations in the toxicity spectrums of various IC combinations. Increased suppression of bone marrow hematopoiesis was the main toxicity while ICIs were combined with conventional chemotherapy. Other adverse events include anaemia, increased transaminase, increased blood bilirubin, hypothyroidism, palmar-plantar erythrodysesthesia, and peripheral sensory neuropathy. The incidence of specific adverse events caused by nivo-ipi was observed to be the lowest in the toxicity spectrums, which may be related to the early termination of enrolment in the nivo-ipi group in CheckMate 649.

### 4.2. Strengths and Comparison with Other Studies

Real-World Evidence (RWE) plays a valuable role in complementing the findings of RCTs and enhancing the generalizability of results. However, Real-World Data (RWD), from which RWE is derived, can often be incomplete, missing, or contain inaccuracies due to variability in clinical practices, patient reporting, or data entry. This lack of rigour and consistency compared to RCT data can introduce biases and confounding variables. Therefore, the current network meta-analysis exclusively incorporates data from RCTs to minimize these issues. This approach ensures greater comparability among various ICIs combined with chemotherapy and allows for a more precise estimation of treatment effects in specific patient populations.

Compared to previously published network meta-analyses [38,39,40,41] evaluating first-line immunotherapies for AGC patients, our analysis possesses the following strengths: Firstly, to minimize heterogeneity and inconsistency, trials recruiting HER-2-positive patients were excluded, and the comparison of each group was based on the same PD-L1 CPS. Then, given the potential for different chemotherapy regimens in individual trials to result in weak transitivity, we undertook categorizations and conducted separate analyses based on the chemotherapy regimen. Unfortunately, only a comparison based on oxaliplatin plus capecitabine was possible. Furthermore, we conducted subgroup analyses stratified by ethnicity, liver metastatic status, primary tumour site, Lauren classification, and microsatellite instability to obtain the most comprehensive results. Finally, a Bayesian framework was utilized to derive a ranking of treatments within each subgroup.

### 4.3. Implications

This systematic review synthesized the evidence from all available and eligible RCTs, providing clinicians with a reference source for assessing the advantages and disadvantages of making practice choices among various promising regimens. A synthesis of recent trial results and most recent guidelines allows for the identification of appropriate treatment modalities for AGC. It is also recommended that a similar approach to data acquisition, study combination clinical practice, and potentially differential subgroup management be adopted in the context of other types of cancer. Further studies should concentrate on the enhancement of anticancer efficacy while simultaneously minimizing the occurrence of adverse effects. Furthermore, it is essential to advance the treatment of gastric cancer in a more individualized manner.

### 4.4. Limitations

The present study was subjected to several limitations. The constraints of existing clinical trials precluded the possibility of eliminating unavoidable confounding factors in this network meta-analysis. Firstly, all IC combinations are compared indirectly by establishing a connection through the node of chemotherapy. Thus, the uniformity of chemotherapy regimens served as the primary means of protecting transitivity. Despite our initial intention to conduct specific analyses based on a homogeneous chemotherapy regimen, the lack of available data prevented us from analysing overall survival in a subset of patients receiving oxaliplatin plus capecitabine. Secondly, funnel plots were not employed to assess publication bias and minor study effects because of the limited number of trials included in each comparison. Thirdly, patients were usually treated with second or later-line therapies, and these subsequent treatments are not uniform, which might hamper the interpretation of overall survival. The heterogeneity observed in the studies may be partly attributed to the use of overall survival as an endpoint to assess the efficacy of each treatment.

## 5. Conclusions

This network meta-analysis indicated that, although it was associated with increased hematopoiesis toxicity, cadonilimab combined with platinum-based chemotherapy was the preferred initial treatment for AGC patients. With acceptable toxicity, IC combinations usually result in a more favourable prognosis when compared to conventional chemotherapy and were preferentially recommended to those with optimistic PD-L1 CPS. These findings have the potential to complement existing guidelines and facilitate more referencing designs of future trials for advanced gastric cancer.

## 6. Registration

The protocol was registered in the Prospective Register of Systematic Reviews (PROSPERO CRD42024560986).

## Figures and Tables

**Figure 1 curroncol-32-00112-f001:**
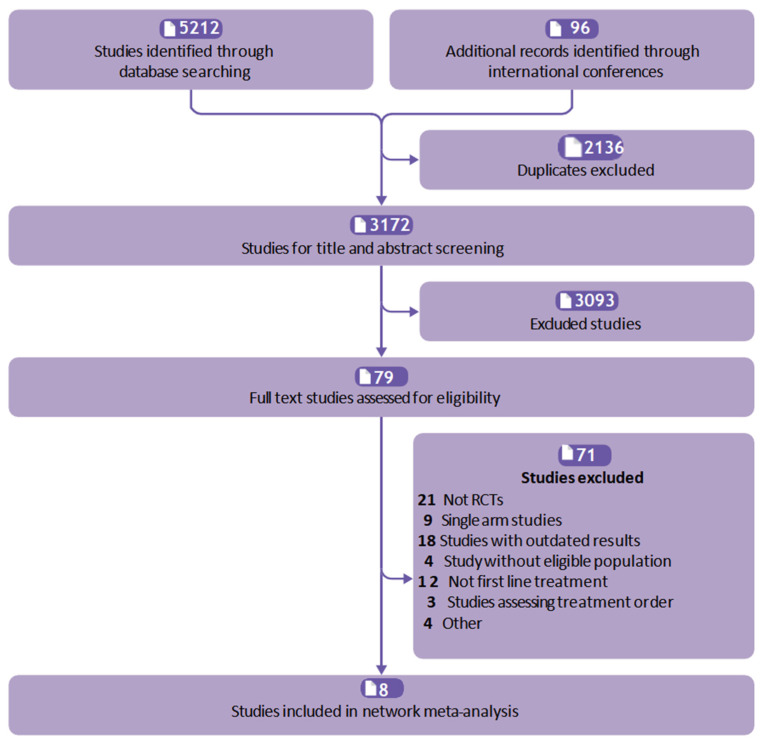
Study selection.

**Figure 2 curroncol-32-00112-f002:**
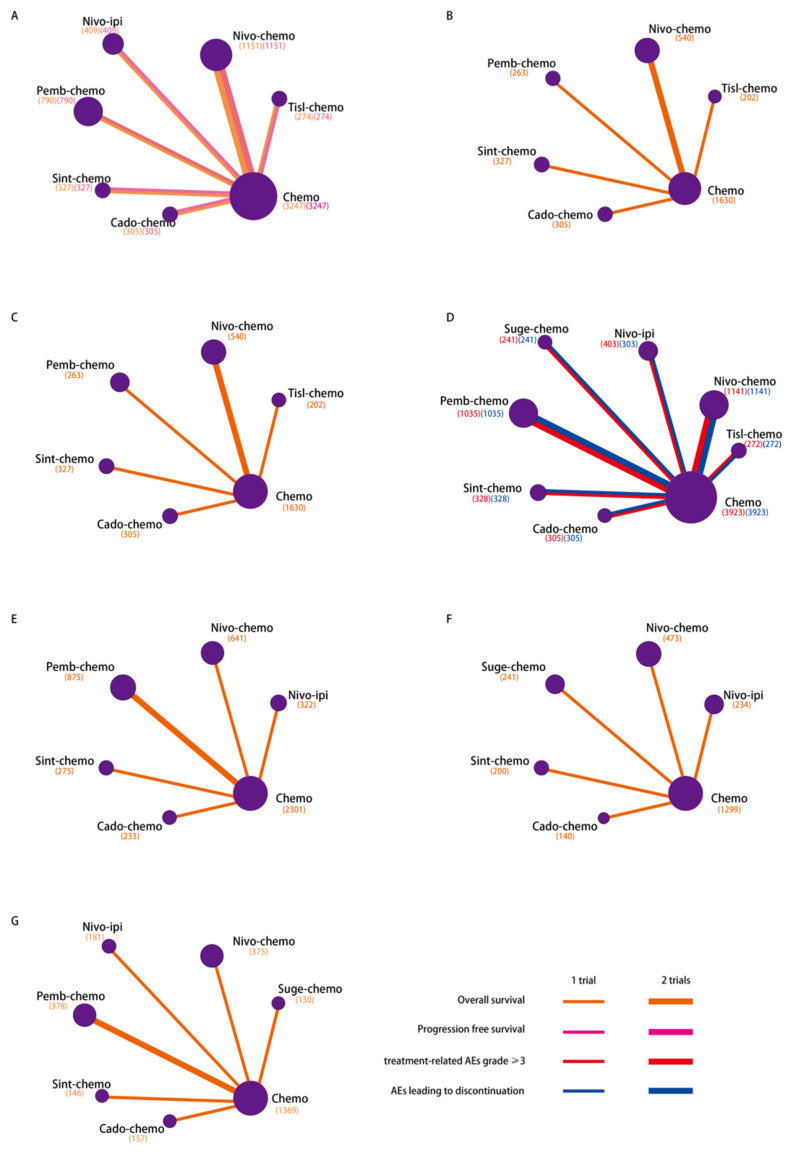
Network diagrams of comparisons on different outcomes of treatments in specific groups. (**A**) Comparisons on overall survival and progression-free survival in ITT population. (**B**) Comparison on overall survival in Asian patients. (**C**) Comparison of overall survival in patients receiving chemotherapy with XELOX. (**D**) Comparisons on TRAEs of grade 3 or higher and AEs leading to discontinuation. (**E**) Comparison of overall survival in PD-L1 CPS ≥ 1. (**F**) Comparison of overall survival in PD-L1 CPS ≥ 5. (**G**) Comparison of overall survival in PD-L1 CPS ≥ 10. Each circular node corresponds to a distinct therapeutic intervention. Node diameter is proportional to the total patient cohort receiving each treatment (numerical values in parentheses). Connecting lines denote direct head-to-head comparisons, with edge width scaled to reflect the number of randomized controlled trials evaluating the linked treatment pairs. TRAE, treatment-related adverse events; ITT: intent-to-treat; tisl-chemo, tislelizumab plus chemotherapy; nivo-chemo, nivolumab plus chemotherapy; nivo-ipi, nivolumab plus ipilimumab; pemb-chemo, pembrolizumab plus chemotherapy; sint-chemo, sintilimab plus chemotherapy; cado-chemo, cadonilimab plus chemotherapy; suge-chemo, sugemalimab plus chemotherapy; chemo, chemotherapy.

**Figure 3 curroncol-32-00112-f003:**
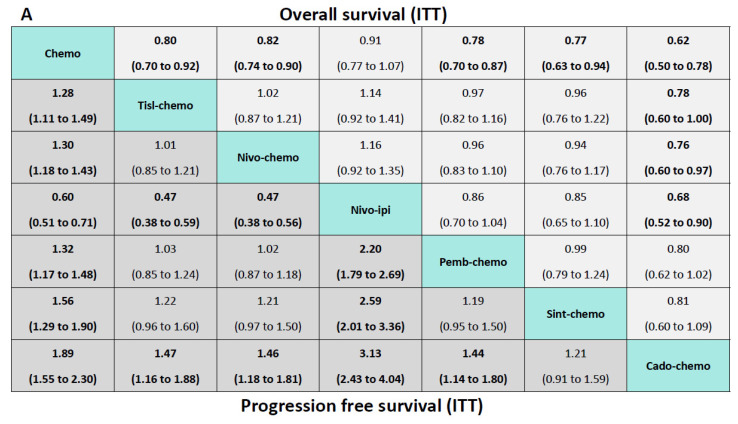
Network meta-analysis pooled outcomes. (**A**) Upper triangular section: hazard ratios (95% CrI) for overall survival (OS); lower triangular section: hazard ratios (95% CrI) for progression-free survival (PFS). (**B**) Upper triangular section: odds ratios (95% CrI) for grade ≥ 3 adverse events; lower triangular section: adverse events leading to discontinuation. (**C**) Upper triangular section: hazard ratios (95% CrI) for overall survival for Asian patients; lower triangular section: hazard ratios (95% CrI) for overall survival for patients receiving chemotherapy with XELOX. (**D**) Upper triangular section: hazard ratios (95% CrI) for overall survival for patients with PD-L1 CPS ≥ 1; lower triangular section: hazard ratios (95% CrI) for overall survival for patients with PD-L1 CPS ≥ 5. (**E**) Upper triangular section: hazard ratios (95% CrI) for overall survival for patients with PD-L1 CPS ≥ 10. The data within each cell represent the hazard ratios or odds ratios (with 95% CrI) for the comparison between the treatment defined by the row and by the column. A hazard ratio of less than 1 and an odds ratio of greater than 1 indicates a favourable effect for the row-defining treatment. Significant results are highlighted in bold. ITT: intent-to-treat; Pemb, pembrolizumab; Tisl-chemo, tislelizumab plus chemotherapy; Nivo-chemo, nivolumab plus chemotherapy; Nivo-ipi, nivolumab plus ipilimumab; Pemb-chemo, pembrolizumab plus chemotherapy; Sint-chemo, sintilimab plus chemotherapy; cado-chemo, cadonilimab plus chemotherapy; Suge-chemo, sugemalimab plus chemotherapy; Chemo, chemotherapy.

**Figure 4 curroncol-32-00112-f004:**
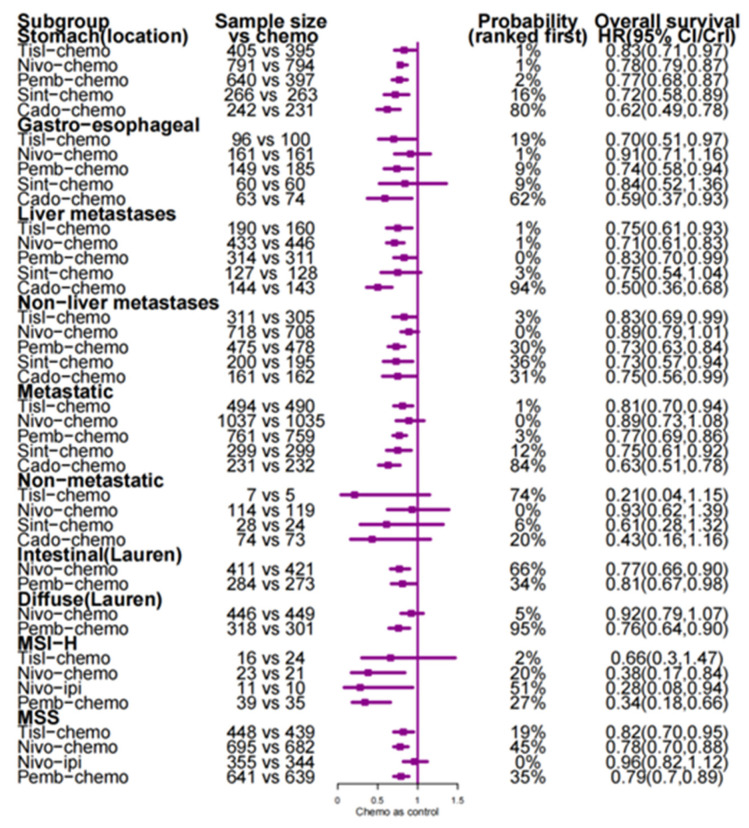
Comparison of overall survival on specific subgroups.

**Figure 5 curroncol-32-00112-f005:**
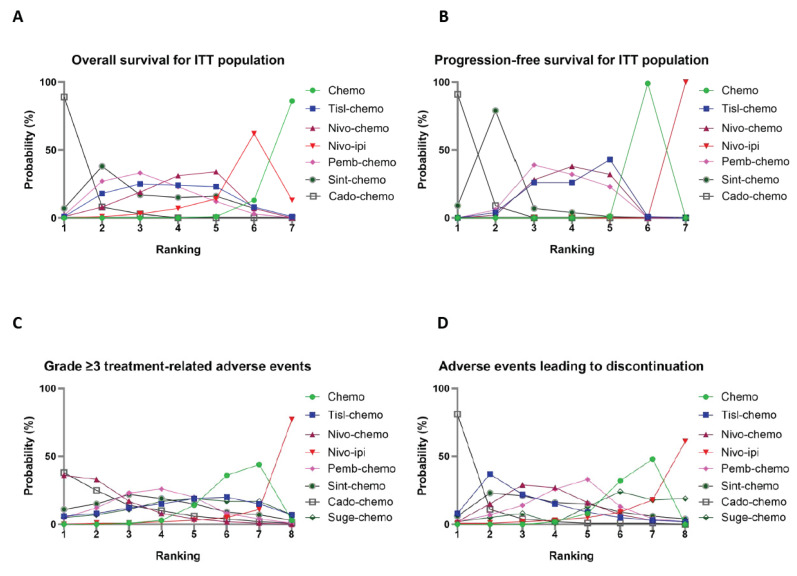
Bayesian ranking profiles of treatments. These profiles depict the probabilities of each comparable treatment being ranked from first to last in terms of overall survival, progression-free survival, adverse events of grade ≥ 3, and adverse events leading to discontinuation of therapy. Ranking curves are described according to the Bayesian ranking results presented in Appendix A. Chemo, chemotherapy; Tisl-chemo, tislelizumab plus chemotherapy; Nivo-chemo, nivolumab plus chemotherapy; Nivo-ipi, nivolumab plus ipilimumab; Pemb-chemo, pembrolizumab plus chemotherapy; Sint-chemo, sintilimab plus chemotherapy; Cado-chemo, cadonilimab plus chemotherapy; Cado-chemo, cadonilimab plus chemotherapy; Suge-chemo, sugemalimab plus chemotherapy.

**Table 1 curroncol-32-00112-t001:** Baseline characteristics of studies included in the network meta-analysis of patients with advanced gastric cancer.

Study; Registered ID	Sample Size (No); Median Age	Phase; Ethnicity	Her-2 Status; Male/Female	PD-L1 Expression Status	Intervention Arm	Control Arm	Reported Outcomes
**RATIONALE 305;** **NCT03777657**	501/496; 61/62	III; multiple	Her-2 negative; NG	ITT (546);	Tisl 200 mg Q3W + chemo (XELOX or FP)	Placebo Q3W + chemo (XELOX or FP)	Overall survival; progression-free survival; grade ≥ 3 TRAEs; AEs leading to discontinuation
**CheckMate 649;** **NCT02872116**	789/792; 62/61	III; multiple	Her-2 negative; 1100/481	ITT (1581); CPS ≥ 1 (1297); CPS ≥ 5 (955); CPS ≥ 10 (767)	Nivo 360 mg Q3W or 240 mg Q2W + chemo (XELOX or FOLFOX)	Chemo (XELOX or FOLFOX)	Overall survival; progression-free survival; grade ≥ 3 TRAEs; AEs leading to discontinuation
	409/404; 62/61		Her-2 negative; 548/255	ITT (813); CPS ≥ 1 (641); CPS ≥ 5 (473); CPS ≥ 10 (379)	Nivo 1 mg/kg + Ipi 3 mg/kg Q3W for 4 cycles, followed by nivo 240 mg Q2W	Chemo (XELOX or FOLFOX)	
**ATTRACTION-4;** **NCT02746796**	362/362; 64/65	III; Asian	Her-2 negative; 523/201	ITT (724)	Nivo 360 mg Q3W + chemo (SOX or XELOX)	Placebo + chemo (SOX or XELOX)	Overall survival; progression-free survival; grade ≥ 3 TRAEs; AEs leading to discontinuation
**KEYNOTE-062;** **NCT02494583**	257/250; 61/62/62	III; multiple	Her-2 negative; 554/209	CPS ≥ 1 (763); CPS ≥ 10 (281)	pemb 200 mg Q3W + chemo (FP)	Placebo plus chemo (FP)	Overall survival; progression-free survival; grade ≥ 3 TRAEs; AEs leading to discontinuation
**KEYNOTE-859;** **NCT03675737**	790/789; 61/62	III; multiple	Her-2 negative; 1071/508	ITT (1579); CPS ≥ 1 (1235); CPS ≥ 10 (551)	Pemb 200 mg Q3W + chemo (XELOX or FP)	Placebo IV Q3W + chemo (XELOX or FP)	Overall survival; progression-free survival; grade ≥ 3 TRAEs
**ORIENT-16;** **NCT03745170**	327/323; 60/60	III; Asian	Her-2 negative; 483/167	ITT (546); CPS ≥ 1 (641); CPS ≥ 5 (397); CPS ≥ 10 (288)	Sint 3 mg/kg for body weight < 60 kg, 200 mg for ≥ 60 kg + chemo (XELOX)	Placebo + chemo (XELOX)	Overall survival; progression-free survival; grade ≥ 3 TRAEs; AEs leading to discontinuation
**COMPASSION-15;** **NCT05008783**	305/305; 63/64	III; Asian	Her-2 negative; 474/143	ITT (710); CPS ≥ 1 (420); CPS ≥ 5 (256); CPS ≥ 10 (153)	Cado 10 mg/kg Q3W + chemo (XELOX)	Placebo + chemo (XELOX)	Overall survival; progression-free survival; grade ≥ 3 TRAEs; AEs leading to discontinuation
**GEMSTONE-303;** **NCT03802591**	241/238 63/63	III; Asian	Unknown 350/129	CPS ≥ 5 (479); CPS ≥ 10 (258)	Suge 1200 mg Q3W + chemo (XELOX)	Placebo + chemo (XELOX)	Overall survival; progression-free survival; grade ≥ 3 TRAEs; AEs leading to discontinuation

Data are expressed as intervention/control unless indicated otherwise. PD-L1: programmed death ligand 1; TRAE: treatment related adverse events; NG: not given; ITT: intent-to-treat; Tisl: tislelizumab; Nivo: nivolumab; Ipi: ipilimumab; Pemb: pembrolizumab; Sint: sintilimab; Cado: cadonilimab; Suge: sugemalimab; Chemo: chemotherapy; XELOX: oxaliplatin 130 mg/m^2^, Day 1 + capecitabine 1000 mg/m^2^ BID Day 1–14, Q3W; FP: cisplatin 80 mg/m^2^, Day 1 + 5-FU 800 mg/m^2^/day, Day 1–5, Q3W; FOLFOX: leucovorin 400 mg/m^2^, day 1 + fluorouracil 400 mg/m^2^, day 1 and 1200 mg/m^2^, days 1–2, and oxaliplatin 85 mg/m^2^, day 1, Q2W; SOX: oxaliplatin 130 mg/m^2^ + S-1 40 mg/m^2^ Q3W once every 3 weeks twice daily on days 1–14, Q3W; Q2W: once every 2 weeks.

## Data Availability

All data was obtained from PubMed, Embase, Web of Science, Cochrane Library, ClinicalTrials.gov databases, and several critical international conferences.

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
