# Peer review of "Immunotherapy Combined with Chemotherapy in the First-Line Treatment of Advanced Gastric Cancer: Systematic Review and Bayesian Network Meta-Analysis Based on Specific PD-L1 CPS"

_curroncol, 2025, doi:10.3390/curroncol32020112_

Round 1

Reviewer 1 Report

Comments and Suggestions for Authors

Thank you for giving me the opportunity to review this paper. Here are my major comments:

The inclusion and exclusion criteria are poorly justified and may introduce bias. Specific concerns include:

- Why were only Phase III trials included?

- Phase II trials with mature survival data could enhance the robustness of the findings.

- Why were HER2-positive patients entirely excluded?

- Many trials report HER2-stratified results, which could be analyzed separately.

- Excluding RWE limits generalizability. The authors should justify this choice.

- What was the cutoff for PD-L1 CPS subgroup inclusion?

- If some trials did not stratify patients by CPS, how were they handled in the analysis?

Heterogenity issue:

- Heterogeneity is high (I² = 67% for PFS, 51.1% for treatment discontinuation), but there is no detailed heterogeneity analysis or justification for how this was handled.

Please give more insights on subgroup analysis

Author Response

Dear Reviewer,

Thank you for dedicating your valuable time to review our manuscript and for your recognition and support of our work. Our team extends our sincere gratitude for your constructive feedback. Below are our point-by-point responses to the comments you raised. Should any inaccuracies remain, we would greatly appreciate your guidance in bringing them to our attention.

Comment/Suggestion 1 and 2: Why were only Phase III trials included? Phase II trials with mature survival data could enhance the robustness of the findings.

Response 1 and 2: We sincerely appreciate your insightful questions and constructive suggestions. We fully concur with your perspective that incorporating phase II clinical trials with mature survival data would strengthen the reliability of our conclusions. During the initial study design phase, we intended to include eligible Phase II/III trials meeting the following criteria:

Population: locally advanced or metastatic gastric cancer patients

Intervention: immunotherapy-chemotherapy combinations vs chemotherapy controls

Setting: first-line treatment

However, during systematic trial screening, we identified that available phase II trials fulfilling inclusion criteria lacked adequately mature outcome data. Additionally, our planned subgroup analyses stratified by PD-L1 expression levels and chemotherapy regimens necessitated robust datasets. Consequently, we ultimately restricted inclusion to phase III trials meeting all predefined criteria. We have compiled a comprehensive list of Phase II clinical trials excluded from the analysis, along with their corresponding exclusion rationales.

  1. The primary endpoint of the study by Hegewisch-Becker S et al. was the objective response rate (ORR), which lacks data on overall survival (OS) and progression-free survival (PFS), and is thus considered to have immature data.

Hegewisch-Becker S, Mendez G, Chao J, Nemecek R, Feeney K, Van Cutsem E, Al-Batran SE, Mansoor W, Maisey N, Pazo Cid R, Burge M, Perez-Callejo D, Hipkin RW, Mukherjee S, Lei M, Tang H, Suryawanshi S, Kelly RJ, Tebbutt NC. First-Line Nivolumab and Relatlimab Plus Chemotherapy for Gastric or Gastroesophageal Junction Adenocarcinoma: The Phase II RELATIVITY-060 Study. J Clin Oncol. 2024 Jun 10;42(17):2080-2093. doi: 10.1200/JCO.23.01636. Epub 2024 May 9. PMID: 38723227; PMCID: PMC11191068.

  1. The study by Bang YJ et al. reported on patients who received platinum and fluorouracil as first-line therapy and had stable disease, partial response, or complete response, and subsequently received ipilimumab as maintenance therapy. The control group received best supportive care (including the prior chemotherapy regimen). Therefore, we excluded this trial.

Bang YJ, Cho JY, Kim YH, Kim JW, Di Bartolomeo M, Ajani JA, Yamaguchi K, Balogh A, Sanchez T, Moehler M. Efficacy of Sequential Ipilimumab Monotherapy versus Best Supportive Care for Unresectable Locally Advanced/Metastatic Gastric or Gastroesophageal Junction Cancer. Clin Cancer Res. 2017 Oct 1;23(19):5671-5678. doi: 10.1158/1078-0432.CCR-17-0025. Epub 2017 Jun 27. PMID: 28655793.

  1. KN046 is a bispecific antibody targeting PD-L1 and CTLA-4. In the study by Shen L et al., KN046 was combined with KN026 as a treatment regimen for patients with HER2-positive disease. This trial was excluded

Shen L, Gong J, Niu Z, Zhao R, Chen L, Liu L, Deng T, Lu L, Zhang Y, Li Z, Li X. 1210P The preliminary efficacy and safety of KN026 combined with KN046 treatment in HER2-positive locally advanced unresectable or metastatic gastric/gastroesophageal junction cancer without prior systemic treatment in a phase II study. Annals of Oncology. 2022 Sep 1;33:S1102.

  1. In the study by Catenacci DV, the primary endpoint for Cohort B is survival; however, no data are currently available for reporting.

Catenacci DV, Rosales MK, Chung HC, Yoon HH, Shen L, Moehler MH, Kang YK. Margetuximab (M) combined with anti-PD-1 (retifanlimab) or anti-PD-1/LAG-3 (tebotelimab)+/-chemotherapy (CTX) in first-line therapy of advanced/metastatic HER2+ gastroesophageal junction (GEJ) or gastric cancer (GC).

  1. The study by Peng Z et al. also has no data results reported yet.

Peng Z, Zhang X, Liang H, Zheng Z, Wang Z, Liu H, Hu J, Sun Y, Zhang Y, Yan H, Tong L. Atezolizumab and trastuzumab plus chemotherapy in patients with HER2+ locally advanced resectable gastric cancer or adenocarcinoma of the gastroesophageal junction: A multicenter, randomized, open-label phase II study.

  1. The study by Al-Batran SE et al. also targets patients with advanced gastric cancer, with modified FOLFOX plus nivolumab and ipilimumab as the experimental group and modified FOLFOX as the control group. Currently, no data have been reported on the results.

Al-Batran SE, Pauligk C, Goetze TO, Riera-Knorrenschild J, Goekkurt E, Angermeier S, Kullmann F, Thuss-Patience PC, Homann N, Ettrich TJ, Junge S. Modified FOLFOX versus modified FOLFOX plus nivolumab and ipilimumab in patients with previously untreated advanced or metastatic adenocarcinoma of the stomach or gastroesophageal junction: Moonlight, a randomized phase 2 trial of the German Gastric Group of the AIO.

Comment/Suggestion 3 and 4: Why were HER2-positive patients entirely excluded? Many trials report HER2-stratified results, which could be analyzed separately.

Response 3 and 4: Thank you for the time and effort you have contributed to improving the quality of our research! The primary objective of this analysis is to guide clinical practice. Referring to the NCCN guidelines as the main basis, it is suggested that for most patients (HER2-negative and MSS), chemotherapy or the combination of chemotherapy and immunotherapy should be used. To align with the guidelines and to fully reflect the concept of precision in treatment population selection, the focus of this study is to compare the efficacy and safety of chemotherapy or various immunotherapy combinations with chemotherapy. Patients were stratified according to PD-L1 expression, region, and the chemotherapy regimen received, and subgroup analyses were conducted based on the primary tumor location, presence of liver metastasis, and Lauren classification.

The NCCN guidelines recommend that patients with HER2 overexpression positivity can opt for chemotherapy plus anti-HER2 therapy plus immunotherapy. However, in current clinical trials, such as the KEYNOTE-811 trial, patients were assigned to receive either pembrolizumab + fluoropyrimidine and platinum-based therapy + trastuzumab or placebo + fluoropyrimidine and platinum-based therapy + trastuzumab. This results in the control group of the trial being placebo + anti-HER2 therapy + chemotherapy, which cannot be incorporated into any node of the network meta-analysis. Other agents, such as the Anti-PD-1/HER2 bispecific antibody IBI315, are currently in the preclinical trial phase, and there are no mature Phase II/III clinical trial data available for reference.

Including cases with definitive HER2 positivity in the scope of this study would make our results more valuable for a broader range of patients, but it might compromise the reliability of the results. Moreover, among the clinical trials included in our analysis, all except the GEMSTONE-303 trial are HER2-negative. Therefore, the main subject of this study is patients with HER2 overexpression negativity. Taking all these factors into account, we excluded clinical trials related to HER2 positivity.

Comment/Suggestion 5: Excluding RWE limits generalizability. The authors should justify this choice.

Response 5: We appreciate your pointing out this issue, and we acknowledge the value of RWE in complementing RCT findings and enhancing generalizability. Real World Data (RWD) can be incomplete, missing, or contain inaccuracies due to variability in clinical practices, patient reporting, or data entry. RWE is derived from RWD, which may lack the rigor and consistency of data from RCTs. The current network meta-analysis only considers data from RCTs rather than RWE to minimize selection bias and confounding variables. This approach ensures higher comparability among various immune checkpoint inhibitors (ICIs) combined with chemotherapy and allows for more precise estimation of treatment effects in specific patient populations.

We have provided a corresponding explanation for this choice in the discussion section of our manuscript, which are highlighted in blue font.

Comment/Suggestion 6: What was the cutoff for PD-L1 CPS subgroup inclusion?

Response 6: Cutoff of PD-L1 CPS

Study

ITT

CPS ≥ 1

CPS ≥ 5

CPS ≥ 10

RATIONALE 305

×

×

×

CheckMate 649

ATTRACTION-4

×

×

×

KEYNOTE-859

×

ORIENT-16

COMPASSION-15

KEYNOTE-062

×

×

GEMSTONE-303

×

×

Dear Reviewer, the cutoff for PD-L1 CPS subgroup inclusion is reflected in Table 1.

Comment/Suggestion 7: If some trials did not stratify patients by CPS, how were they handled in the analysis?

Response 7: During the process of including trials in our analysis, we found that the RATIONALE 305 trial stratified patients based on the Tumor Area Positivity Score (TAP), while the ORIENT-16 trial not only used the Combined Positive Score (CPS) but also stratified patients using the Tumor Cell Proportion Score (TPS). Other trials utilized the CPS method. Given the differences in PD-L1 expression scoring methods, forcibly incorporating populations from various cohorts into the network meta-analysis may disrupt baseline consistency, thereby compromising the reliability of the findings. Therefore, we have included only the intent-to-treat population from the RATIONALE 305 trial in this analysis. For ORIENT-16 and other eligible trials, we used data stratified according to PD-L1 CPS.

Response 8: Heterogeneity is high (I²= 67% for PFS, 51.1% for treatment discontinuation), but there is no detailed heterogeneity analysis or justification for how this was handled.

Reply: Dear Reviewer,

We sincerely appreciate your insightful comments and constructive suggestions. Upon conducting a systematic review of the included trials, the heterogeneity arising from trial design and statistical methodologies was deemed negligible. The most plausible source was attributed to random variation within the dataset. Accordingly, we implemented a leave-one-out approach to iteratively screen potential outliers (Figure S8), integrated with heterogeneity assessment outcomes (Figure S4). This process identified the COMPASSION-15 trial as the primary contributor to the elevated heterogeneity in the current analysis.

Consequently, a second sensitivity analysis was performed after excluding this trial. The revised results have been incorporated into the "Sensitivity Analysis" section of the main text (highlighted in blue font for clarity), with additional supporting materials appended to the supplementary files (Figures S9-S11).

Comment/Suggestion 9: Please give more insights on subgroup analysis.

Reply: Dear Reviewer,

Thank you for your valuable suggestions for revision. After conducting subgroup analyses based on conditions such as Lauren classification, we have derived the probabilities of treatment rankings within each subgroup, as well as other relevant data. Given the constraints on the word count and the number of figures in the manuscript, we have chosen to present the results of the subgroup analyses in the form of forest plots to ensure conciseness. We have also recognized that our initial description of the subgroup analysis results was not sufficiently clear. Therefore, we have revised the description of the subgroup analysis results and highlighted these revisions in blue font within the text. The specific revisions are as follows:

Our analytical results consistently demonstrated that regardless of whether the primary tumor was located in the stomach or at the gastroesophageal junction, cado-chemo exhibited the highest probability of being ranked first in therapeutic efficacy. In addition to cado-chemo, sint-chemo may confer greater therapeutic benefit in gastric tumors, while tisl-chemo demonstrates more pronounced efficacy in gastroesophageal junction tumors. In patients with liver metastasis, cado-chemo demonstrated the highest likelihood of first-line therapeutic superiority, whereas sint-chemo appeared to confer a comparative therapeutic advantage in cases without liver involvement. We conducted stratified analyses based on gastric cancer disease status. The results demonstrated that cado-chemo remains the most clinically advantageous regimen for metastatic gastric cancer, while revealing no statistically significant difference between IC combinations and chemotherapy alone in locally advanced disease. In the stratified analysis based on Lauren classification, nivo-chemo provided greater benefits than pembrolizumab combined with chemotherapy in gastric cancer of the intestinal type. Conversely, the analysis results for gastric cancer of the diffuse type were opposite. There have been many literature reports on the relationship between the microsatellite instability status and the application of immune checkpoint inhibitors30,31. Gastric cancer of the MSI-H type represents only a small fraction of all cases, yet our results demonstrated excellent responsiveness to immune combination therapy, particularly the nivo-ipi. In contrast, for MSS tumors, analysis had shown that the nivo-ipi did not provide as much benefit as the IC combination.

Thank you once again for your valuable contributions to improving the quality of this study! Given that we are required to submit the revised manuscript within a specified timeframe, please do not hesitate to point out any areas where our work may fall short. We wish you smooth sailing in your work and a pleasant life!

Reviewer 2 Report

Comments and Suggestions for Authors

The work is timely and highly relevant, addressing the challenges of modern therapy for advanced cancer. Congratulations to the authors for their effort. They provide an analysis of studies evaluating the statistical outcomes of combining immunotherapies with standard chemotherapy for advanced gastric cancer.

Content Observations

  1. Stratification of Cases

The results would gain greater precision if cases were stratified between locally advanced cancer and metastatic cancer.

  1. Evaluation of Chemotherapy

Statistical evaluation should focus exclusively on cases with a single chemotherapy line, without including patients who received other oncological treatment lines.

  1. Complex Stratification

The selection and guidance toward a specific therapy require a more complex stratification. Factors such as tumor location, recurrent disease, Lauren classification, microsatellite instability, and tumor markers should have been included. These would allow for a more coherent direction for patients toward appropriate treatments.

Formatting Remarks

  1. Bibliographic References

The format of the references is inadequate and needs revision.

  1. Figure 5

The resolution of Figure 5 is very low. It is recommended to replace it with a higher-quality version.

Author Response

Dear Reviewer,

Thank you for dedicating your valuable time to review our manuscript and for your recognition and support of our work. Our team extends our sincere gratitude for your constructive feedback. Below are our point-by-point responses to the comments you raised. Should any inaccuracies remain, we would greatly appreciate your guidance in bringing them to our attention

Comment/Suggestion 1: Stratification of Cases   

The results would gain greater precision if cases were stratified between locally advanced cancer and metastatic cancer.

Response 1: Thank you very much for your comments. Your suggestions have provided new insights that helped refocus our research priorities. Stratifying the disease status into locally advanced and metastatic gastric cancer can significantly improve the accuracy of our conclusions. We conducted subgroup analyses categorizing disease status as metastatic and non-metastatic, presenting both the probability of each treatment regimen being ranked first and the corresponding hazard ratios in forest plots. Compared to alternative visual representations such as network diagrams, forest plots offer greater space efficiency while simplifying the presentation of results. Should you deem it necessary, we would be happy to revise the manuscript using a network meta-analysis approach.

Comment/Suggestion 2: Evaluation of Chemotherapy

Statistical evaluation should focus exclusively on cases with a single chemotherapy line, without including patients who received other oncological treatment lines

Response 2: Thank you for your insightful comments. We fully agree with your suggestion that the evaluation of therapeutic efficacy should focus exclusively on first-line treatment regimens, as subsequent lines of therapy may confound the interpretation of outcomes. In clinical research, post-progression treatment selection (e.g., second- or third-line therapies) is highly heterogeneous due to the complexity of drug resistance, and these subsequent interventions can dilute the observed overall survival results, making it challenging to objectively assess the true efficacy of the initial treatment. Therefore, stratified analyses of overall survival and progression-free survival are both essential to ensure the reliability of conclusions.

In our initial analysis design, we planned to stratify patients by various factors (e.g., PD-L1 expression levels, tumor location, disease status, Lauren classification) and analyze OS and PFS separately within each subgroup. However, upon detailed review of clinical trial publications, we noted that many trials did not report PFS data for these stratified subgroups. Given this limitation, we ultimately opted to analyze both OS and PFS exclusively in the intent-to-treat population.

We acknowledge that the absence of granular subgroup data represents a limitation of this study, and we sincerely appreciate your constructive feedback. Should you require further clarification or additional analyses, please do not hesitate to let us know.

Comment/Suggestion 3: Complex Stratification

The selection and guidance toward a specific therapy require a more complex stratification. Factors such as tumor location, recurrent disease, Lauren classification, microsatellite instability, and tumor markers should have been included. These would allow for a more coherent direction for patients toward appropriate treatments.

Response 3: We sincerely appreciate the time and effort you have dedicated to improving the quality of our study. Stratifying patients based on complex factors such as primary tumor location, Lauren classification, and microsatellite status can indeed enhance the precision of therapeutic strategies. However, due to space constraints, we have included only the forest plots for these subgroups in the manuscript, along with the probabilities of each treatment regimen being ranked first in terms of efficacy.

If you strongly recommend a more detailed analysis of these factors, we will prioritize revising the manuscript accordingly at the earliest opportunity.

Comment/Suggestion 4: Bibliographic References

The format of the references is inadequate and needs revision.

Response 4: Thank you for pointing out the issue. We have revised the references in accordance with the requirements provided by the editorial office.

Comment/Suggestion 5: The resolution of Figure 5 is very low. It is recommended to replace it with a higher-quality version.

Response 5: Thank you for your feedback! We have adjusted the resolution of Figure 5 as per your suggestion. When you view and zoom in on this image, you will be able to see the content clearly.

Thank you once again for your valuable contributions to improving the quality of this study! Given that we are required to submit the revised manuscript within a specified timeframe, please do not hesitate to point out any areas where our work may fall short. We wish you smooth sailing in your work and a pleasant life!

Round 2

Reviewer 1 Report

Comments and Suggestions for Authors

The authors have satisfied my suggestions.

Kind regards